

# Derivation of bedrock topography measurement requirements for the reduction of uncertainty in ice sheet model projections of Thwaites Glacier

Blake Castleman[1,2], Nicole-Jeanne Schlegel[1], Lambert Caron[1], Eric Larour[1], Ala Khazendar[1]

[1]Jet Propulsion Laboratory, California Institute of Technology, Pasadena, CA, USA
[2]Georgia Institute of Technology, Atlanta, GA, USA

*Correspondence to*: Blake A. Castleman (bcastleman3@gatech.edu)

**Abstract.** Determining the future evolution of the Antarctic Ice Sheet is critical for understanding and narrowing the large existing uncertainties in century-scale global mean sea level rise (SLR) projections. One of the most significant glaciers/ice streams in Antarctica, Thwaites Glacier, is at risk of destabilization and, if destabilized, has the potential to be the largest regional-scale contributor of SLR on Earth. This is because Thwaites Glacier is vulnerable to the marine ice sheet instability, as its grounding line is significantly influenced by ocean-driven basal melting rates, and its bedrock topography retrogrades into kilometer deep troughs. In this study, we investigate how bedrock topography features influence the grounding line migration beneath Thwaites Glacier when extreme ocean-driven basal melt rates are applied. Specifically, we design experiments using the Ice-Sheet and Sea-level System Model (ISSM) to quantify the SLR projection uncertainty due to reported errors in the current bedrock topography maps that are often used by ice sheet models. We find that spread in model estimates of sea level rise contribution from Thwaites glacier due to the reported bedrock topography error could be as large as 21.9 cm after 200 years of extreme ocean warming. Next, we perturb the bedrock topography beneath Thwaites Glacier using wavelet decomposition techniques to introduce realistic noise (within error). We explore the model space with multiple realizations of noise to quantify what spatial and vertical resolutions in bedrock topography are required to minimize the uncertainty in our 200-year experiment. We conclude that at least a 2 km spatial and 8 m vertical resolution would independently constrain possible SLR to ±2 cm over 200 years, fulfilling requirements outlined by the 2017 Decadal Survey for Earth Science. Lastly, we perform an ensemble of simulations to determine in which regions our model of Thwaites Glacier is most sensitive to perturbations in bedrock topography. Our results suggest that the retreat of the grounding line is most sensitive to bedrock topography in proximity to the grounding line's initial position. Additionally, we find that the location and amplitude of the bedrock perturbation is more significant than its sharpness and shape. Overall, these findings inform and benchmark observational requirements for future missions that will measure ice sheet bedrock topography, not only in the case of Thwaites Glacier but for Antarctica on the continental scale.



## 1 Introduction

The future of the West Antarctic Ice Sheet (WAIS) is known to be one of the largest sources of uncertainties in global mean sea level rise (SLR) on a century time scale (Schlegel et al., 2018; Yu et al., 2019). Its inherent instability results from much of its bedrock being below sea level (Fretwell et al., 2013) and its proximity to warm ocean temperatures (Schodlok et al., 2016). Continued acceleration of interior retreat of its grounding line and the loss of ice volume above floatation (VAF) could tip WAIS towards irreversible collapse (Rignot et al., 2014; Milillo et al., 2019). For these reasons, it is critical to understand

the regional sensitivities in order to project future changes in the glacier. To characterize how the behavior of various ice shelves throughout WAIS may evolve in the future, the cryosphere community relies heavily on numerical modeling and simulations. Such tools allow physically-based prediction and quantification of the sensitivity of grounding line evolution. This aids the community in determining which regional factors and characteristics most significantly contribute to the local stability of individual glaciers.


Within WAIS, Thwaites Glacier is a highly dynamic and sensitive region of the ice sheet (Robel et al., 2019; Rignot, 2001). Unstable and vulnerable to damage (Lhermitte et al., 2020), this glacier holds the potential for a ~.59 m global mean SLR and may initiate the collapse of WAIS (Holt et al., 2006). The glacier is held by pinning points present in undersea mountain ridges around its grounding line (Seroussi et al., 2017). Within 100 km upstream of the grounding line, Thwaites Glacier's bedrock

slope becomes retrograde, splitting into trenches more than a kilometer in depth. As a result, its grounding line is considered marginally stable, presently between states of equilibrium and retreat (Payne et al., 2004; Seroussi et al., 2017).

Past model-based experiments suggest that two of the strongest controlling mechanisms for Thwaites Glacier are ocean-driven basal melting rates and bedrock topography (Larour et al., 2019; Nias et. al., 2016; Schlegel et al., 2018; Waibel et. al., 2018).

For basal melting rates, because stochastic evolution is prevalent in ocean circulation, especially beneath ice shelves, melting rates are difficult to accurately observe, model, and predict. Temporal variability in ocean forcing further amplifies uncertainty in ocean-driven melt rates and predictions of future ice sheet responses (Khazendar et al., 2019; Robel et al., 2019). Ocean circulation models physically estimate the dynamic evolution of the ocean-induced basal ice shelf melt rates. Though this can estimate melt rates at the higher spatial and temporal resolutions needed to inform ice sheet models, accuracy can only be

ensured up to the present-day grounding line. Once the current configuration evolves, extrapolation is required, meaning more uncertainty is contributed to the system (Marshall and Clarke, 1997; Nakayama et al., 2019; Zhou and Hattermann, 2020). If dynamic model coupling between the ice and the ocean is used, larger sensitivity issues can arise and amplify uncertainties as they are propagated through a model simulation (Klerk and Voormeeren, 2008). As a result, a number of more simplistic models have been proposed, such as calculating constant basal melting rates based on incoming ocean heat convection (Rignot

et al., 2016; Holland and Jenkins, 1999; Bondzio et al., 2018). Thermal forcing can also be used as functions of ice shelf depth due to ocean salinity and temperature gradients (Yu et al., 2018). Fundamentally, it is nearly impossible to represent all oceanic





processes occurring using simplified parameterizations. Indeed, recent model-based studies suggest that century-scale uncertainty in SLR potential under different basal melting rate scenarios has a significant spread (Yu et al., 2018; Schlegel et al., 2018). Together, this all suggests that uncertainty in estimates of ocean forcing may remain a significant source (and

perhaps the largest) of uncertainty in ice sheet model simulations of Thwaites Glacier into the near future.

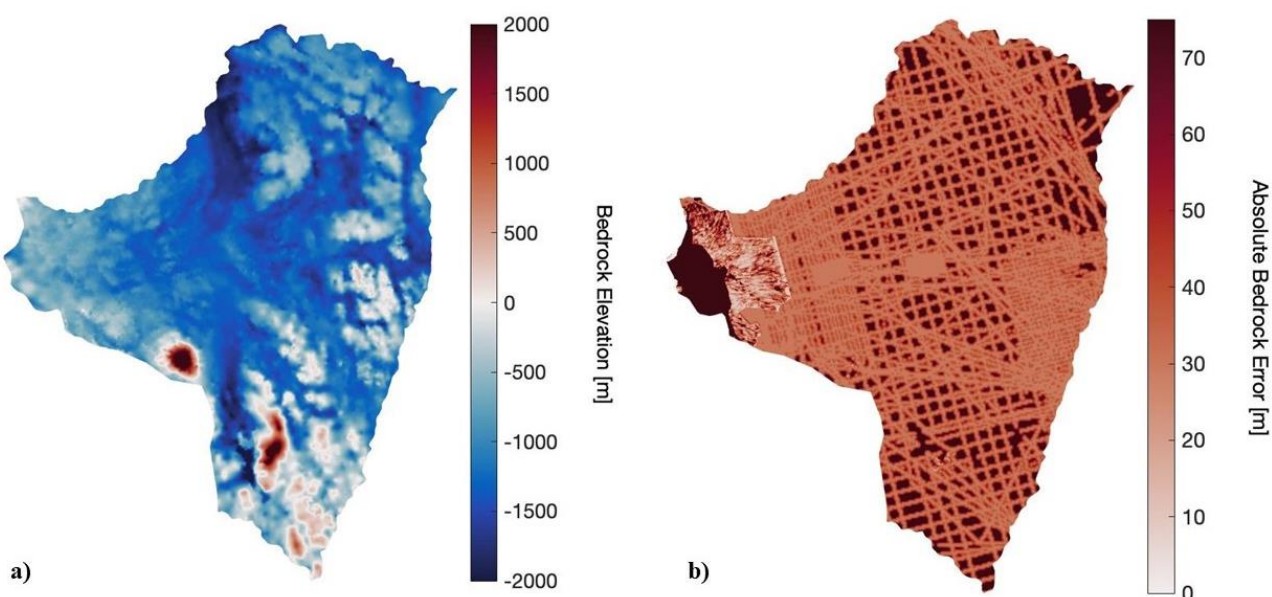

**Figure 1: (a)** Thwaites bedrock topography from BedMachine (Morlighem et al., 2019) is given with **(b)** its corresponding
uncertainty (considered to represent 3σ) in the region. The data is interpolated onto our ISSM domain (later detailed) from its
original grid map.

Bedrock topography, however, is more quantifiable, as it is fixed in time on decadal time-scales (i.e., aside from longer-term

processes such as glacial isostatic adjustment and erosion/sedimentation) and is directly measurable using remote sensing,
namely radar altimetry. Decades of remote sensing data from past missions have been used to construct estimates of present-
day underlying bedrock topography. This has resulted in generations of digital elevation models (DEM) with varying degrees
of coverage and accuracy around Antarctica. The measurements however are based on radar tracks and thus physically-
informed interpolation procedures are needed to fill in the spatial gaps between measurements (e.g., mass conservation,

Morlighem et al., 2019). However, even these state-of-the-art products are plagued by statistical errors especially in areas far
from the radar tracks. Consequently, even the most informed estimates of bedrock topography are associated with significant
error in regions where measurements are sparse (Fig. 1).



In this study, we design a suite of ice sheet model experiments to investigate how known errors in bedrock topography affect

200-year simulations of the response of Thwaites Glacier to an extreme increase in ocean-driven basal melt rates. To accomplish this, we first perturb the domain bedrock within error to its minimum (bed minus error) and maximum (bed plus error) possible bedrock configurations. Using these new boundary conditions for simple sensitivity experiments, we characterize the spread in modeled SLR contribution resulting from error in bedrock topography (Sect. 3). Next, we further investigate model sensitivity to bedrock error using wavelet techniques to perturb the bedrock. In this case, the experiments

are designed to derive the spatial and vertical resolutions needed to reduce uncertainty in modeled estimates of the glacier's future evolution within the requirements outlined by the 2017 Decadal Survey for Earth Science (Sect. 4). Finally, we perform an uncertainty quantification (UQ) sampling experiment to simulate probabilistic model outcomes followed by a sensitivity test to locate regions where the bedrock topography of Thwaites plays the most significant role in determining grounding line evolution (Sect. 6).

## 95 2 Model Setup

### 2.1 Model Description

We use the Ice-Sheet and Sea-level System Model (ISSM; Larour et al., 2012), a thermomechanical high resolution ice sheet model, to simulate the forward transient evolution of Thwaites Glacier. Our regional model is originally derived by extracting

the Thwaites basin from an ISSM model of the Antarctic Ice Sheet (Schlegel et al., 2018; Seroussi et al., 2020). Here, the model uses initial surface and bedrock topography ($\bar{x}$) from BedMachine v2 DEM (Morlighem et al., 2019). Our sample space of possible bedrock realizations ($\pm3\sigma$) is derived from the absolute error provided with this product, which we consider to be equivalent to a 95% confidence bound (or $3\sigma$). Initialization and relaxation of our model follows the procedures described for the JPL_ISSM model by Seroussi et al. (2020), including the determination of the basal friction coefficient over grounded ice

and the ice viscosity of the floating ice using data assimilation techniques (Morlighem et al., 2010) to best match observed velocities (Rignot et al., 2011a; Rignot et al, 2017). For computational efficiency, all simulations are run with a 2-dimensional Shelfy stream stress balance approximation (SSA; MacAyeal, 1989), after Schlegel et al. (2018) (See Sect. 2.3 below). For other detailed information about the model parameterizations and setup, including the treatment of basal friction and the rheology law, we refer the reader to Schlegel et al. (2018).

### 110 2.2 Mesh and Boundary Conditions Initialization

The upstream ice boundaries of the regional Thwaites Glacier domain are initially determined by the continental-model ice divides and are then modified following Yu et al., 2018. At these boundaries, all thickness and velocity values are held constant as single point constraints during all simulations and are specified by the thickness and velocity values of the continental model





of Schlegel et al. (2018) (Schlegel et al., 2013). At the calving front, a free-flux condition is imposed, and the ice front positions
are held constant, based on the mask from Morlighem et al., 2019. The grounding line evolves according to a sub-element
evolution scheme and assumes hydrostatic equilibrium (Seroussi and Morlighem, 2018).

To create new high-resolution meshes adequate for our sensitivity studies to capture spatial variability in the BedMachine
product, we use ISSM's static anisotropic mesh adaptation, informed by the gradient in initial surface velocities. Here, we
define two different meshes, one for our vertical test (VT) experiments, with horizontal resolution defined between 200 m to
1000 m and the other for our spatial test (ST) experiments, with horizontal resolution defined between 50 m to 200 m (Fig.
S1). Due to the large amount of finely resolved elements in the ST model, we further reduce the domain in the upstream interior
regions where ice velocities and thickness changes are found to not be significantly perturbed during our most extensive
Thwaites grounding line retreat scenarios. This results in large reductions in computational costs. The VT model contains
about four hundred thousand elements while the ST model contains about 1.3 million elements. Because the mesh resolutions
vary between spatial and vertical models, SLR results slightly differ between their control simulations. This result agrees with
Seroussi and Morlighem, 2018, who conclude that higher resolution models result in more SLR as compared to lower
resolution models of the same setup (especially resolution drops between 500 m to 2 km). Note that we observe this divergence
primarily in cases where the grounding line migration rate is the largest, specifically when the glacier has already committed
to full collapse at sumulation year 150.

The friction and ice viscosity are taken from the continental model JPL_ISSM described by Seroussi et al. (2020) and the
surface mass balance and ocean basal forcing are after Schlegel et al. (2018). We interpolate these values onto our new meshes
using bicubic interpolation, and after Schlegel et al. (2018), they are held constant through all simulations.

**2.3 Stress Balance Approximation**

All forward simulations conducted here make use of the SSA stress balance approximation, chosen for its computational
efficiency, which advantageously decreases the computational costs of running the large number of simulations required by
our analysis. In addition, sensitivity experiments conducted on our VT model suggest that the use of a higher order
approximation (Blatter, 1995; Pattyn, 2003) does not significantly affect model results. These results agree with Schlegel et
al., 2018, who state that a two-layer thin-film stress balance approximation (Schoof and Hindmarsh, 2010; Hindmarsh, 2004)
has a statistically insignificant effect on grounding line sensitivity to perturbations in model boundary conditions with respect
to SSA. Furthermore, we assume the effects of thermal variations are slow relative to the grounding line processes modeled
here. After running our own sensitivity tests, we have confirmed that they do not significantly affect our results on the 200-
year timescale investigated. Therefore, in the absence of a thermal model, a three-dimensional representation of the mesh is
not required. As a result, for the purposes of this investigation, we take SSA as an acceptable approximation for stress balance.





We set the stress balance mechanical equilibrium residual convergence criterion to 1% and the stress balance velocity relative convergence criterion to .1%, both tested under decreased convergence criteria to ensure repeatability, stability, and robustness.

## 2.4 Grounding Line Migration and Forcing

Basal melting rates are calculated using the Massachusetts Institute of Technology General Circulation Model (MITgcm;
Marshall and Clarke, 1997) using techniques described in Schodlok et al. (2016) and implemented after Schlegel et al. (2018). That is, basal melt rates near the grounding line are extended inland using a nearest neighbor extrapolation method. This results in an aggressive basal warming from oceanic heat flux into newly formed ice shelf cavities as the grounding line retreats. Furthermore, we include a multiplier of ×1.8 on the melt rates in order to account for the maximum melting rate realistically achievable within the basin, in the case that Antarctic Bottom Water were to intrude underneath the ice shelf (Schlegel et al,
2018). This choice allows our experiments to explore the domain's full SLR contribution potential over the simulation period of 200 years.

The grounding line migration and grounding line friction interpolation are set using a sub-element on partially floating elements scheme. The melt interpolation is set to have no melting on partially floating elements, which results in a conservative estimate
of sea-level contribution compared to using a sub-element melting scheme. We choose this option, because past studies have shown that the use of no melt on partially floating elements produces realistic results when spatial resolution needs to be compromised for computational efficiency. That is, using the no melt scheme, the modeled grounding line behaviour more closely matches that of simulations using much more highly-resolved spatial mesh in proximity to the grounding line (Seroussi and Morlighem, 2018).


We obtain an initial grounding line position by calculating where the BedMachine ice thickness mask provided has buoyant forces exceeding gravitational forces. Then, before applying the ×1.8 basal melt multiplier, we let the model run with its control bedrock (bedrock topography realization given by BedMachine) for 10 years to relax it and stabilize the grounding line position (Schlegel et al., 2018; Seroussi et al, 2011; Gillet-Chaulet et al., 2012). Comparing our grounding line position to the initial
grounding line position in Yu et al (2018), we observe that at worst we are overshooting the initial grounding line by approximately 10 km at the largest offsets (Fig. S2).

Before running a model simulation forward in time, we follow an algorithm on the glacier geometry within ISSM to ensure that the simulation does not experience a shock to the mass transport, stress balance, and grounding line solutions after bedrock
perturbation is performed. These details are documented in the Appendix (Sect. 9.1).

Models are run on two Broadwell nodes with 28 cores each (56 processors total) on the Pleiades Supercomputer Cluster using ISSM version 4.16. Runs are set for 200 years using a timestep of about ~6.1 days, or 60 timesteps per year.



## 3 Experiment 1: Minimum and Maximum Bedrock Resulting Spreads

To begin our investigation, we run a simple experiment on our 200-year forward simulation to test model SLR contribution sensitivity within our defined bedrock sampling space. That is, we run two simulations, one using the minimum and one using the maximum possible bedrock topography ($\bar{x}$-3$\sigma$ and $\bar{x}$+3$\sigma$), where 3$\sigma$ represents the map of bedrock error reported by BedMachine. Together, the results of this experiment bound the possible SLR contributions (calculated from the change in VAF from grounding line migration) and final grounding line positions that our forward simulation could yield when forced

with variations of bedrock topography within error. The results of these experiments, presented in Fig, 2., are a simple first-order illustration of the magnitude of decadal-scale sea-level projection uncertainty sourced in present day Thwaites bedrock error.

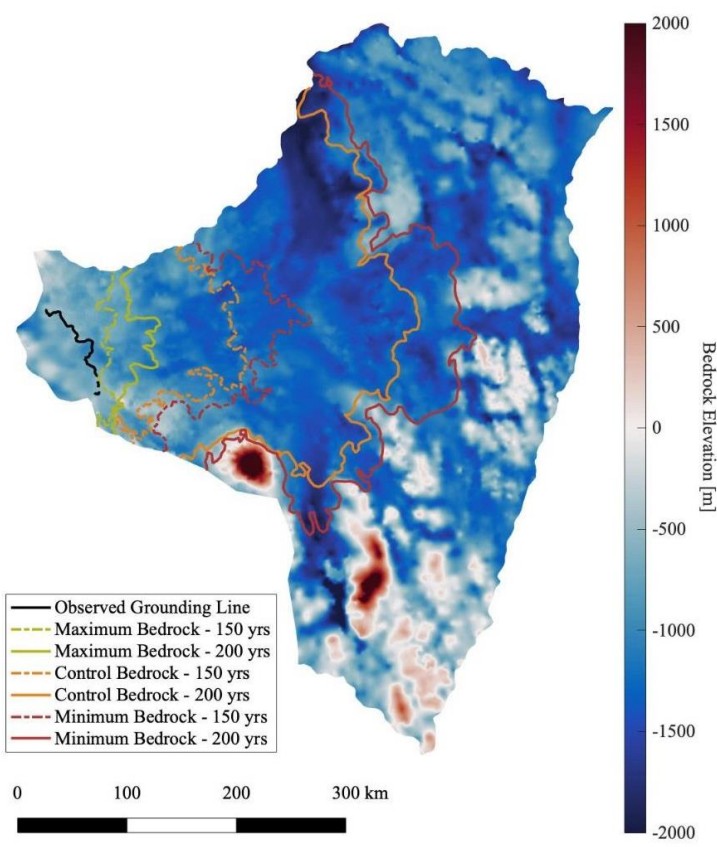


**Figure 2**: Results of forward model experiments using the control (orange), maximum (yellow), and minimum (red) bedrock configurations are shown. Two timestamps are depicted: 150 years (dashed) and 200 years (solid). At 200 years, the control, maximum, and minimum bedrock configuration models yield 21.1 cm, 4.8 cm, and 26.7 cm of global mean SLR



respectively. The resulting SLR difference between the models of minimum and maximum grounding line retreat yield 21.9

195        cm. See the Appendix (Sect. 9.3) for the 150 yr and 200 yr tabulated SLR values.

The results presented in Fig. 2 suggest that under the maximum bedrock scenario, stabilization can occur within the first 50 km of the current grounding line position (Fig. S3, ridge set (a)). In contrast, the ridge farther upstream (Fig. S3, noted as ridge set (b)) is not capable of stabilizing Thwaites under our extreme warming conditions, instead it merely slows the widespread

retreat. These results support conclusions posed by Morlighem et al. (2020) that continuous retreat in Thwaites occurs (given our modeling assumptions) if the glacier retreats past its initial ridges (Fig. 2a in Morlighem et al. (2020); Fig. S3, noted as ridge set (a)), especially under an extreme ocean warming condition. Since the majority of the bedrock upstream of these initial ridges is more than a kilometer deep, the presence of significant pinning point features is indeed improbable, even within the sampling space of large errors that exist in the region. Overall, we conclude that the current error in bedrock topography is

responsible for a global mean SLR difference of 21.9 cm (maximum SLR - minimum SLR) within 200 years, under a forcing of extreme ocean warming.

We also find that the control bed topography simulation itself (without perturbation) forced with extreme basal melt rates results in significant retreat in a 200-year period (Fig, 2); the bedrock elevation is too low for any pinning points to create

resistance. The maximum bedrock scenario (as compared to the other models), on the other hand, illustrates that our model results are highly sensitive to the bedrock error. We see here that the grounding line migrates minimally, only about 50km upstream over our 200-year forcing period. Indeed, we find that the grounding line evolution and consequential sea-level contribution can be significantly altered (between 21.1 cm to 4.8 cm) simply through an increase in the control bed topography within error. Therefore, within possible realizations of bed topography considered here, unobserved features or realistic noise

not captured in our control bedrock may constitute distinct pinning points that could stabilize or delay retreat. This means that it is possible for not-yet-observed bedrock features that exist within the current bedrock error bounds to play critical roles in dictating the future retreat rate of Thwaites Glacier.

It is also important to note that results suggest that there is little difference in retreat and SLR contribution between the

minimum and control bedrock topography scenarios. This is due to model sensitivity to the extreme (×1.8) basal melt rates, which leads to the ungrounding of most of the ice in the model domain for both simulations; in other words, the minimum scenario has retreated to higher ridges deep within the domain such there are no more opportunities for further retreat and SLR. The grounding lines for both runs migrate through Thwaites Glacier's deep trenches over the simulation period, resulting in significant loss of ice VAF and a SLR contribution close to the domain's maximum potential.


The results show that the two extremes bedrock scenarios (maximum vs. minimum) diverge significantly in a 200-year period, suggesting that Thwaites is highly sensitive to bedrock perturbations given the simulations' aggressive basal melt forcing.



Therefore, we find that, within error, perturbations in bedrock topography are capable of slowing grounding line retreat in response to extreme ice shelf melt rates, and we cannot assume that the current vertical error in bedrock topography is
negligible within the Thwaites basin. Consequently, our results suggest that in order to obtain high confidence in robust ice sheet model projections under an aggressive ocean warming scenario, the existing bedrock error bounds within Thwaites Glacier must be reduced.

## 4 Investigating Resolution Requirements for Uncertainty Minimization

Our initial results show that a large projected SLR uncertainty exists with consideration to the currently reported bedrock error.
In other words, our forward simulations of Thwaites Glacier evolution are sensitive to perturbations in bedrock topography. We now ask, if the current error ranges are not sufficient, what bedrock uncertainty range  would be required to accurately model Thwaites Glacier evolution, and at what spatial resolution.

Since an elevation increase or decrease of the entire bedrock topography is not a realistic representation of the effects of error
on the reported BedMachine topographical representation, we use discrete wavelet decomposition (Sect. 4.1) in order to create noise amplifications in our bedrock topography array. Two dimensional discrete wavelet decomposition involves using discrete signals of differing shapes and sizes in order to decompose an array into four subarrays: the An array (low-frequency approximation), $H_n$ array (horizontal high-frequency), $V_n$ array (vertical high-frequency), and $D_n$ array (diagonal high-frequency).

We amplify the high-frequency filters ($H_n$,  $V_n$, and  $D_n$) so that, upon recomposition of the subarrays, we introduce realistic physical noise into our experiments. Because high frequency filters have large coefficients in locations with high frequencies, many of the pinning points (ridges, mountains, etc; Fig. S3, noted as ridge set (a)) that are capable of slowing grounding line retreat in Thwaites Glacier are, by design, targeted by our wavelet image processing. By focusing on these present pinning
points, our analysis is more likely to result in a wide range of resultant model SLR contribution and grounding line retreat.

Here, we design two sets of experiments to derive spatial resolution and vertical error requirements for ice sheet model bedrock topography. The goal of these experiments is to determine what these independent requirements would need to be in order to lower the SLR uncertainty of our forward simulations to ±2 cm. We choose ±2 cm of SLR based on the 2017-2027 Decadal
Survey for Earth Science and Applications from Space (National Academies of Sciences, 2018) which states that it is one of the most important science applications to "quantify the rates of sea-level change and its driving processes at global, regional, and local scales, with uncertainty <0.1 mm/yr" (Chapter 3, S-3a). If we assume a maximum ±.1 mm/yr of SLR uncertainty, such a requirement accumulated over 200 years is equivalent to ±2 cm of SLR uncertainty.





We acknowledge that this is a strict constraint for an ice sheet model projection system, and that our results will represent stringent ideal high-end requirements. Nonetheless, we consider it a meaningful benchmark for the quantification of uncertainty in regional projections of glacial contribution to sea level. Following such, spatial and vertical resolution requirements are quantified throughout the proceeding sections (Sect. 4.2 and Sect. 4.3) as they relate to restricting SLR uncertainty within the ±2 cm range.

**4.1 Wavelet Decomposition Setup**

We use discrete wavelet transform (DWT) for a two-dimensional decomposition and recomposition of bedrock topography. Wavelets form basis functions for projection that reveal the waveform (or frequency content) of the signal around a given location. The method relies on first multiplying the signal by an oscillating function (the wavelet) that vanishes away from the location at which it is applied, and then integrating that product over space. The result is a coefficient that measures the

similarity between the signal and wavelet around that point. The wavelet can be spatially rescaled to analyze the signal at different wavelengths, hence it is widely used as a tool to study the space-frequency content of a signal. Here, we also use the filtering capability of wavelets to generate new realizations of the bedrock topography used as input for our ice model runs.

DWT specifically operates using a pair of wavelets labeled the lowpass and highpass filters which are combined in two

dimensions and used to convolve the signal into 4 sets of coefficients: the approximation coefficients ($A_n$; dual lowpass) which represent a low frequency (smoothed) version of the signal, and the 3 detail coefficients according to their spatial orientation, i.e. x-axis coefficients ($H_n$; highpass x-axis, lowpass y-axis), y-axis coefficients ($V_n$; lowpass x-axis, highpass y-axis), and diagonal coefficients ($D_n$; dual highpass). The wavelet decomposition can in turn be reapplied using rescaled wavelets on the resulting $A_n$ matrix to provide a new set of detail coefficients $H_{n+1}$, $V_{n+1}$, and $D_{n+1}$ of lower frequency than $H_n$, $V_n$, and $D_n$, and

a new approximation matrix $A_{n+1}$. This iterative process is used to characterize the signal at different decomposition/resolution levels here indexed with n (Daubechies, 1997; Mallat, 1989; Meyer, 1995).

We use the second Daubechies wavelet (hereafter referred as db2; Daubechies, 1992) and the discretized Meyer wavelet (hereafter referred as dmey; Meyer, 1990; Abry, 1997) for our noise study. We decide to use these wavelet bases due to their

opposing strengths and weaknesses (Abry, 1997; Daubechies, 1997). The db2 wavelet can localize signals well in the space domain but poorly in the frequency domain while the dmey wavelet localizes signals poorly in the space domain but well in the frequency domain. Therefore, they jointly analyze the full effect of perturbation on our DEM, helping observe the SLR sensitivity of Thwaites in both space and frequency domains.

Using these wavelets to obtain a post-decomposition state, we introduce a multiplier to the $H_n$, $V_n$, and $D_n$ coefficient matrices in order to amplify existing bedrock topography noise. Therefore, upon recomposition of the original image ($A_{n-1}$) using the





$A_n$ approximated coefficients, we obtain a similar bedrock topography that we previously had with the addition of introduced noise on areas with existing high frequency features.

In some cases, we further decompose the $A_n$ coefficient matrix using the same wavelet in order to achieve a multiresolution analysis. Upon higher level decompositions ($n \geq 2$), the high frequency features targeted change to be of larger spatial scale due to single pixels now representing larger areas. Therefore, noise amplification of solely the n layer at these large orders produce geological landforms such as entire ridges and trenches (Daubechies, 1997; Mallat, 1989; Meyer, 1995). We take advantage of this mathematical notion to develop a function to decompose our DEM to an $n$ level, amplify the $H_n$, $V_n$, and $D_n$
coefficient matrices, and then recompose accordingly.

We choose not to conserve ice volume when creating the bedrock realizations in order to simplify the wavelet amplification process. We also maintain the ice surface DEM in places above floatation (see Sect. 9.1 for situations in which the ice surface DEM may be altered). As a result, more divergent results are expected between different bedrock realizations as some models
may have more or less bedrock added at critical pinning points. Sensitivity tests suggest that this divergence, however, is minimal and does not affect our results.

**4.2 Experiment 2: Spatial Resolution Degradation Results**

To understand the spatial resolution required for modeling Thwaites within our given regional domain, we use a mesh that captures all features present in the BedMachine DEM; that is, the ST mesh has a higher spatial resolution (200 m, Sect. 2.2) than BedMachine (500 m).
To determine the bedrock topography and the bedrock error for the ST mesh, we use bicubic interpolation. Using such a finely resolved mesh affords us a bedrock topography that we can manipulate at a higher resolution in order to test the impact of noise below the current 500m spatial resolution.

We begin by upgrading the bedrock realization to a 200 m resolution, amplifying the high frequency filter noise (through
techniques and variations described in Sect. 4.1), and then degrading the resolution to 400 m. We use this 400 m resolution as the control resolution for each specific wavelet-resolution combination. A 400 m spatial resolution is chosen as our control resolution since, at this resolution, a single feature (around 400 m) will be described by multiple mesh elements. At higher resolutions than this, we would only be representing features close to a 200 m spatial resolution with a single element which shocks the system upon degradation. Next, image degradation of the DEM resolution to various spatial resolutions ranging
from 400 m to 30 km follows through bicubic interpolation. Results of SLR contribution difference from the control run at year 200 for the 258 forward simulations for this experiment are shown in Fig. 3.

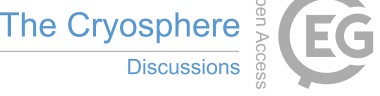

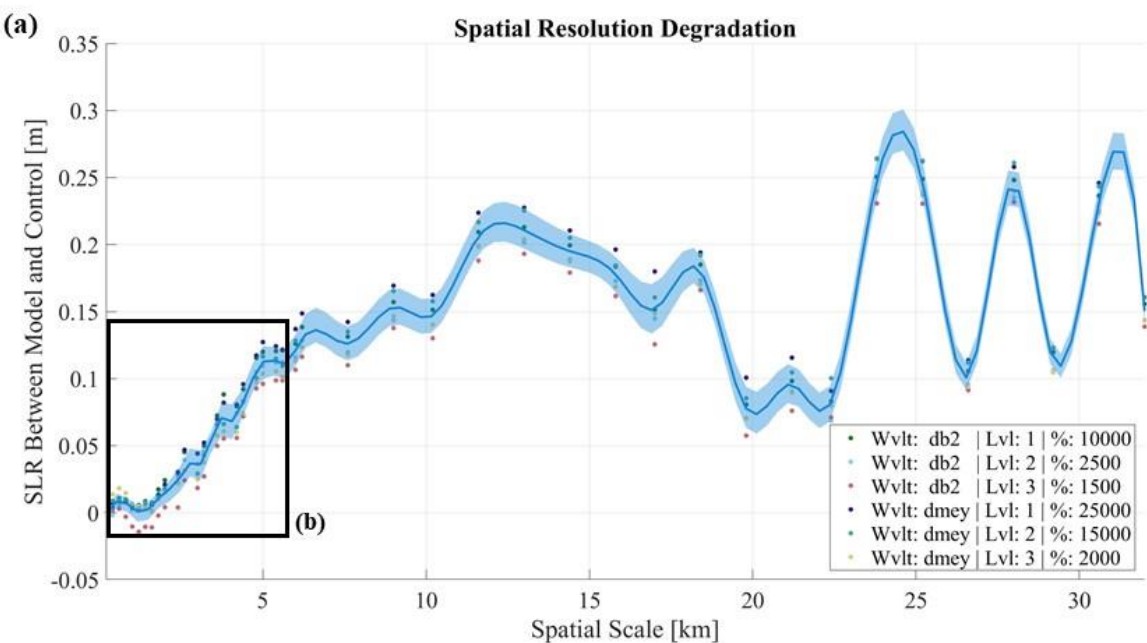

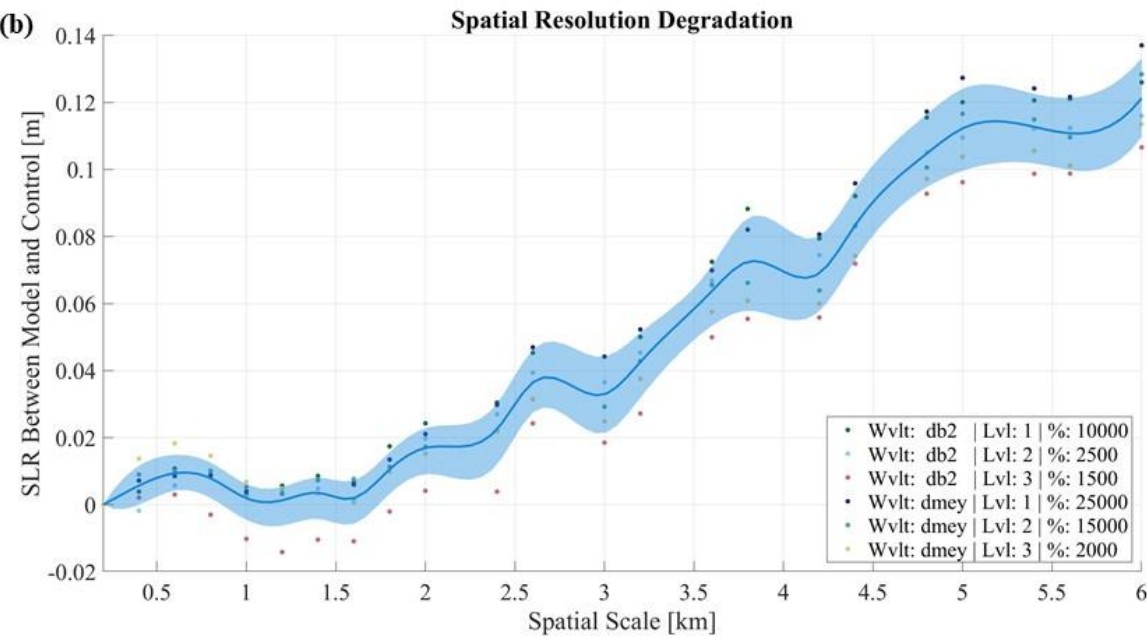

**Figure 3:** Final SLR is plotted against the current downgraded resolution for various noisey bedrock topography (**a**, results for all spatial scales, **b**, results for spatial scales of 6 km and finer). The model results (dots) are spline interpolated with trends depicted with a mean resultant line (dark blue line) and a percentile band (shaded region) of $\pm 1\sigma$ (68.3%). Observing the Appendix (Sect. 9.2) will provide further detail on bedrock perturbation codes from the legend.



In Fig. 3b, we find that results converge under 2 km for all of the constructed maps. Within the first 2 km, much of the positive high frequency noise perturbations about the realization (previous wavelet noise amplifications performed at a 200m level) are either amplified or reduced from elements due to the degradation of resolution (resultant from spatial interpolation). Therefore, relatively minor changes in SLR are observed as compared to the respective control runs due to the compensation of bedrock about these elements. However, after passing the ~2 km threshold, much of the noise is completely lost by the wide grids used

in the interpolation, and after this 2 km threshold, we observe a large sudden gain in SLR difference.

In Fig. 3a, we find that results lack an overall general trend, with random changes in SLR resulting as the resolution is degraded, and no obvious pattern observed. This results from changes in bedrock feature characteristics (e.g. ridge omission, mountain expansion) due to our degradation of spatial resolution (accomplished through spatial interpolation) which leads to highly

unrealistic topography as it crosses the bedrock error range in BedMachine (only when the spatial degradation, resultant from spatial interpolation, occurs at extremely low resolutions). We observe that the SLR difference between each realization's control run and the degraded realization eventually exceeds 2 cm around the 2 km resolution mark, meaning that at resolutions above this threshold the high frequency perturbations that we created previously are lost to interpolation approximations. Since we observe a clear compensation before 2 km, while beyond 2 km they are no longer compensated for, we conclude that the

spatial degradation threshold for proper accuracy lies around this point.

With consideration to our ±2 cm of SLR requirement, we evaluate our spatial resolution limit to be approximately 2 km, particularly at Thwaites' pinning points, as SLR does not approach the 2 cm threshold again for any of the lower spatial resolutions investigated here (Fig. 3a).

**4.3 Experiment 3: Vertical Resolution Noise Construction Results**

To derive vertical resolution constraints in Thwaites, we design a set of experiments to systematically perturb our bedrock topography with noise of increasing amplitudes. We take BedMachine's bedrock topography to be our control model and then use wavelet decomposition to amplify the high frequency noise (similar to Sect. 4.2). Using either the db2 or dmey wavelet in combination with four decomposition levels, we create eight bedrock topography realizations for the model. We then take the

absolute value of the noise created for each realization, resulting in perturbations that are all positive. This allows us to apply a noise realization as either a positive (through addition) or a negative (through subtraction) set of perturbations and then to examine the impact of both types of noise on the model simulations independently, as opposed to having to deconvolve the impact of the two types of perturbations in combination.

To test the effect of vertical changes in bedrock topography, we establish a set of limitations on the maximum height to which new perturbation features can be built. For each mesh vertex in our model bedrock topography map, the bedrock altitude is



equal to the minimum of a) the perturbed bedrock realization, b) the maximum feature height change (Fig. 4, x-axis), and c) the bedrock error limit ($\bar{x}$-3σ and $\bar{x}$+3σ for negative and positive noise respectively) on that vertex. For our experiments, we systematically change the amplitude of our bed topography noise by varying (b) the maximum feature height change between

10m to 100m with 10m increments, giving a total of 20 final bedrock topography realizations per wavelet-decomposition level combination after both positive and negative noise are included, for a total of 160 model simulations.

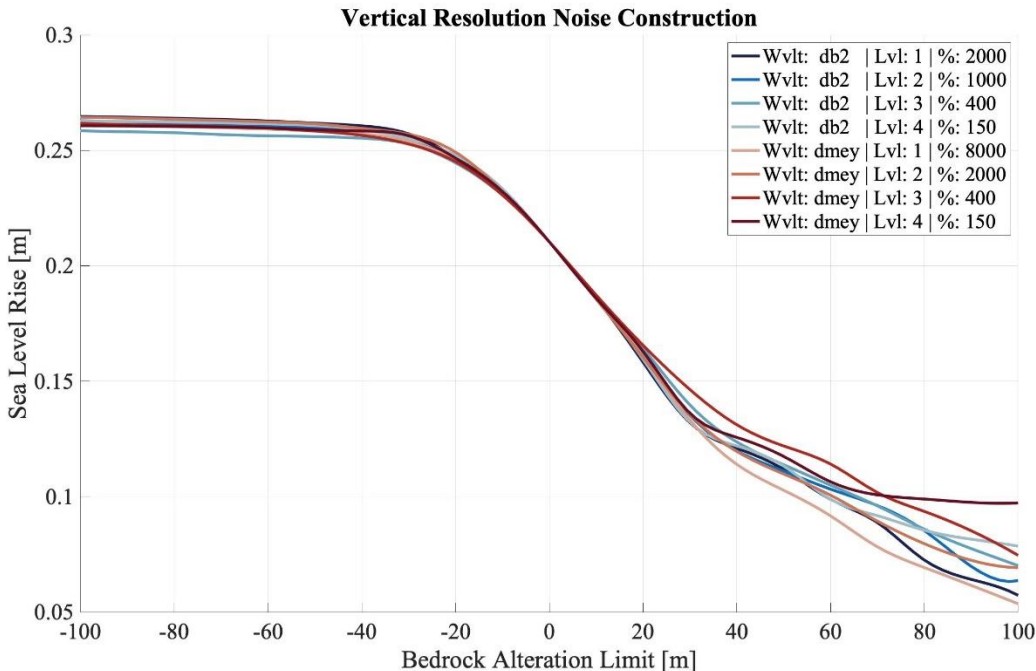

**Figure 4:** Final SLR plotted against the maximum bedrock change threshold designated for each model simulation. Points represent the model results connected using a spline interpolation. Eight different bedrock topographic realizations are made with high-frequency amplification for this experiment. See Appendix (Sect. 9.2) for further detail on bedrock perturbation codes from the legend.

In Fig. 4, we present results from all forward simulation experiments for the vertical resolution tests. We find that the results are nearly linear within ±20 m of bedrock addition with asymptotic behavior dominating with increasing max height perturbation for both the positive and negative addition of noise. The results are biased towards the bedrock error due to different bedrock realizations from various perturbations being leveled off by (c) the bedrock error limit of each pixel.



For negative bedrock changes (bedrock subtraction), the asymptotic behavior observed from about -100 m to -20 m is due to the simulation reaching the limit of maximum glacial retreat. At this point in the modeled evolution of the grounding line, a complete Thwaites destabilization and retreat is achieved, meaning that there is little ice left in the domain that can unground into deep bedrock troughs. Therefore, SLR potential beyond this point is limited, since our SLR calculation is calculated from changes in VAF. Between -20 m to 0 m of bedrock subtraction however, the SLR contribution behaves more linearly, as full

retreat has not yet occured by the time these model simulations have completed.

For positive bedrock changes (bedrock addition), we still observe the destabilization of Thwaites between about 0 m to 20 m of maximum bedrock addition. Less retreat has occurred by the time 200 years has passed since the topographically higher bedrock realizations create a delay in the retreat rate of the grounding line. There are not sufficient pinning points present in

these simulations to completely prevent retreat. Between 20 m to 100 m, however, the glacier obtains much more effective stabilization due to the topographically higher bedrock levels. We find these results to be the only ones where the different wavelet-decomposition level combinations diverge from one another. As a result, there is almost 5 cm of SLR difference between the least and most stable models at 100 meters of maximum bedrock addition.

Our results from these experiments strongly suggest that amplitude of bedrock change matters more than the shape of the perturbation itself. There is little variance between the differing runs and significant agreement in terms of general trend, despite our application of different wavelet shapes within the ensemble. As the two types of wavelet shapes are used at multiple resolutions, no shape seems to have a significantly larger effect on our overall model results. Moreover, because different volumes of bedrock are added to each run, one would also expect larger divergences from the general trend due to alteration

of the available VAF for contribution to SLR. However, we find that this effect is negligible and that the small spread between the various simulations at the same bedrock alteration limit only lends further support that it is the amplitude of change, rather than the type of wavelet, which more heavily determines simulation SLR contribution.

With respect to our ±2 cm of SLR requirement, we focus on the control run at 0 m of bedrock addition, where within Fig. 4,

the tangential slope is the largest. From this point, we observe that vertical bedrock perturbation of a magnitude within approximately ±8 m at areas where the model is most sensitive to bedrock would hold the simulation SLR uncertainty within our target range. It is important to note that, in comparison to variations in spatial resolution, we find that vertical perturbation in bedrock topography contributes more significantly to uncertainty in ice sheet model estimates of SLR contribution from Thwaites Glacier, especially since even a meter of perturbed vertical bedrock elevation has the potential to alter model results

significantly.





## 5 ISSM-DAKOTA Framework

For further investigation of the sensitivity of model simulations to topographic data accuracy in Thwaites, we take advantage of the Design Analysis Kit for Optimization and Terascale Applications (DAKOTA) software from Sandia National Laboratories (Eldred et al., 2008) that has been embedded within the ISSM framework (Larour et al., 2012b; Larour and Schlegel, 2016). DAKOTA, a tool for UQ analysis and statistical error quantification, has traditionally been used to perturb ISSM model input and boundary conditions within various regional domains in order to perform a variety of sensitivity tests on model output diagnostics, like mass flux (Schlegel et al., 2013, 2015; Schlegel and Larour, 2019) and regional mass balance (Schlegel et al., 2018).

In this study, we first use the DAKOTA software to create a statistical sampling of bed error perturbations in order to isolate bedrock pinning points. Here, we launch an ensemble of forward models, applying the predetermined set of perturbations to the bed using the initialization procedure detailed in Appendix (Sect. 9.1). We treat the entire domain as one single partition such that the bedrock everywhere is perturbed by the same percent error for any given ensemble member (Schlegel et al., 2018). The perturbation set consists of a normal sampling distribution of 300 samples within a ±3% bedrock standard deviation, the mean BedMachine standard error determined for the Thwaites basin based on the error supplied by Morlighem et al. (2019). We use Latin hypercube sampling (LHS) algorithm (Swiler and Wyss, 2004) to generate the sampling distribution, after Schlegel et al. (2015).

This first sampling experiment informs us at which locations, under various bedrock configurations within the reported BedMachine standard error (i.e., $3\sigma$), the grounding line prefers to reside. Next, we use these results to isolate the most influential region of bed topography within our domain. We then design an additional spatial sensitivity sampling experiment to determine which specific features within that region have local errors that would influence Thwaites grounding line stability the most. To do so, we partition the targeted region into 400 equal-area subsections using the Chaco software (Hendrickson and Leland, 1995), after Schlegel et al. (2013). We then perturb the bedrock in each partition one at a time by adding the maximum local BedMachine standard error for that partition (Morlighem et al, 2019), and finally run a forward transient, resulting in a total of 400 simulations. Comparison of the grounding line behavior between these runs allows us to identify pinning points by quantifying which individual bedrock regions (partitions) have the largest effect on modeled sea-level contribution when perturbed within error.

### 5.1 Experiment 4: UQ Probabilistic Distribution Results

As previously discussed in Sect. 5, we take advantage of the DAKOTA software in order to understand the probability of divergence from our control model. More specifically, we use DAKOTA functionality to derive a normally-distributed perturbation set, with a $3\sigma$ sampling range of about ±9%, equivalent to the mean uncertainty of BedMachine error within the



Thwaites basin. The perturbation set consists of 300 values, by which we multiply our control bedrock topography over the entire model domain. Each of these new bedrock realizations are run with the basal melt rate multiplier of ×1.8 and a multiplier

of ×1.4, resulting in a total of 600 forward simulations.

In order to compare our results to other SLR drivers, we also use DAKOTA to derive a uniform sampling of ice shelf basal melting rate multipliers ranging between ×1 to ×1.8, in order to represent a full range of possible future ocean-warming scenarios, after Schlegel et al. (2018). We repeat the bedrock sampling experiments, but this time with perturbation of only the

ice shelf basal melting rates. In this way, we are able to compare the impact of errors in bedrock topography against that of uncertainty in ice shelf basal melting rates, which have been found to contribute a significant amount of uncertainty to decadal-scale simulations of the Antarctic Ice Sheet (Schlegel et al., 2018).

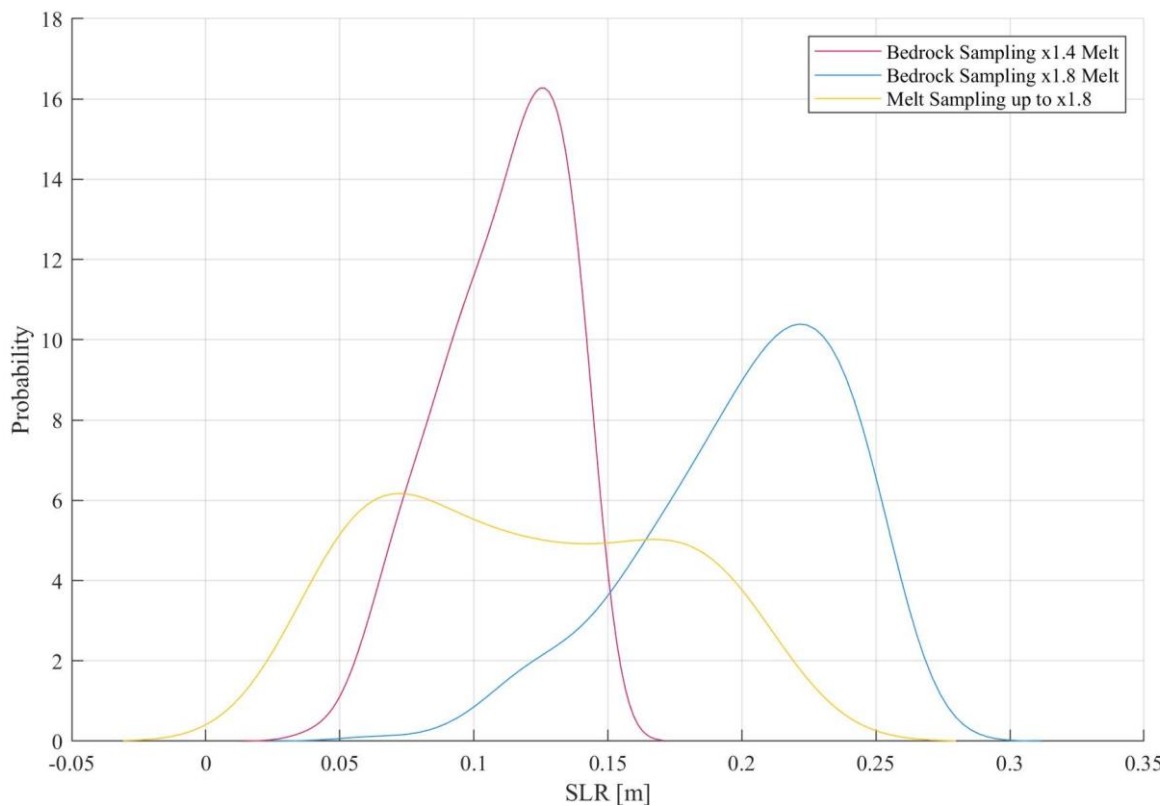


**Figure 5:** Distributions of event probability as a function of SLR contribution after 200 years of simulation for three different boundary condition sampling experiments, each experiment consisting of 300 model simulations. Experiments include sampling of bedrock topography under two different ocean warming scenarios: one model scenario with a basal melt





multiplier of ×1.4 (pink) and one model scenario with a basal melt multiplier of ×1.8 (blue). For comparison, we also present
460       results from sampling of basal melt rates between ×1 to ×1.8, with no changes in bedrock topography (yellow).

We find that under the control bedrock and variable basal melt rates, results exhibit a bimodal distribution between approximately 4 cm to 21 cm of SLR contribution. This suggests that if no error is present in bedrock topography, the second set of ridges inland (Fig. S3, noted as ridge set (b)) can still play an important role in stopping, or delaying, Thwaites' grounding

line retreat on the 200-year timescale investigated here. The uncertainty range due to basal melting rates is comparable to the simulation uncertainty in SLR resulting from bedrock variation under our most extreme ocean warming scenario (Fig. 5, ×1.8 Melt). We find that under this extreme forcing and without alteration to bedrock topography full grounding line retreat is almost achieved, and under lower-end basal melt forcing, i.e. present-day basal melt rates, stability is maintained. The resulting distribution curve, however, is not normal, and its bimodal nature suggests that melt rates must exceed a specific threshold to

ensure a SLR contribution of about 15 cm or greater. The bimodal response also implies that the melt multiplier threshold is dependent on the location of bedrock pinning points beyond Fig. S3, ridge set (a) that must be surpassed in order to achieve full retreat (i.e., Fig. S4).

Under the maximum basal melting rate (×1.8) and a variable bedrock configuration, Thwaites achieves a 21 cm distribution

between about 6 cm to 27 cm of SLR. This large spread in SLR contribution after 200 years of simulation suggests that our model simulations are highly sensitive to the currently reported bedrock error. Furthermore, this distribution's SLR peak probability has a larger magnitude than that of our control SLR contribution. This implies that the grounding line retreat under our most extreme ocean forcing scenario is inherently unstable. Previous results (Fig. 4) suggest that the negative skew of the curve and tail with lower SLR contributions is likely a response to positive bedrock perturbations. Specifically, the addition of

bedrock at influential pinning points promotes glacier stability. Since the sensitivity of modeled grounding line retreat rates is also highly dependent on ocean forcing (i.e., basal melt rates), as discussed above, it is clear that model results are strongly driven by a combination of bedrock topography configuration and ocean-driven floating ice melt rates. This is illustrated in Fig. 5 by bedrock sampling performed with the lower melt multiplier of ×1.4. The use of this multiplier results in a range of SLR contributions that span the area between the two peaks of the melt sampling's distribution and results in a smaller

simulation uncertainty than exhibited by the other two curves. This ranges between about 4 cm to 14.5 cm of SLR contribution.

## 5.2 Experiment 5: Pinning Point Sensitivity Test

To isolate the pinning points and understand their weight on retreat, we design a spatial sensitivity test to determine which areas of bedrock topography are most significant in determining the resultant SLR. Due to the large number of elements within

our domain, it is not suitable to perform the sensitivity test over the entire domain as there is a large computational cost. Therefore, we use the distribution results from the bedrock sampling with ×1.8 melt (Fig. 5) to determine where the most



influential grounding line pinning points may be, or in other words, at what model mesh vertices the grounding line prefers to rest during the 300 bedrock sampling simulations.

Taking grounding line positions at 10-year intervals for all 300 simulations, we calculate the percent time that the grounding line falls within 100m of each mesh vertex (Fig. S4). We take a threshold of 20% or above to denote locations of strong topographical influence on the grounding line, or pinning points. We appropriate this percentage by examining the sensitivity of expected SLR contribution to perturbations in bedrock at locations above various values of percent thresholds. Specifically, we conduct an experiment, in which we add the maximum bedrock error ($3\sigma$) to each mesh vertex above a minimum percent

threshold, and then run a set of simulations varying the threshold value. Results suggest that a threshold of 20% would capture the majority of the influential bedrock pinning points (Fig. S5).

Because all of the resulting points of interest are in proximity to the initial grounding line, we isolate a single subdomain in which to perturb the domain's bedrock topography for our sensitivity test. We decide on 25 km$^2$ for the spatial resolution,

yielding 400 partitions total and requiring a total of 400 forward simulations. Ideally, we would conduct this study using a spatial resolution of about 4 km$^2$ (~2km spatial resolution, Sect. 4.2). 25 km$^2$ is chosen instead however because it drastically decreases the number of partitions, the required number of simulations (one per partition), and therefore the computational price of the sensitivity test, lowering the computation cost by over 75%. Because we expect our sensitivity experiment to consider three independent variables: bedrock error, mean feature altitude, and spatial location, we decide to perturb the bed

by the maximum bedrock error. Choosing a constant perturbation change or a ratio instead would not have fully taken all three of these variables into account.





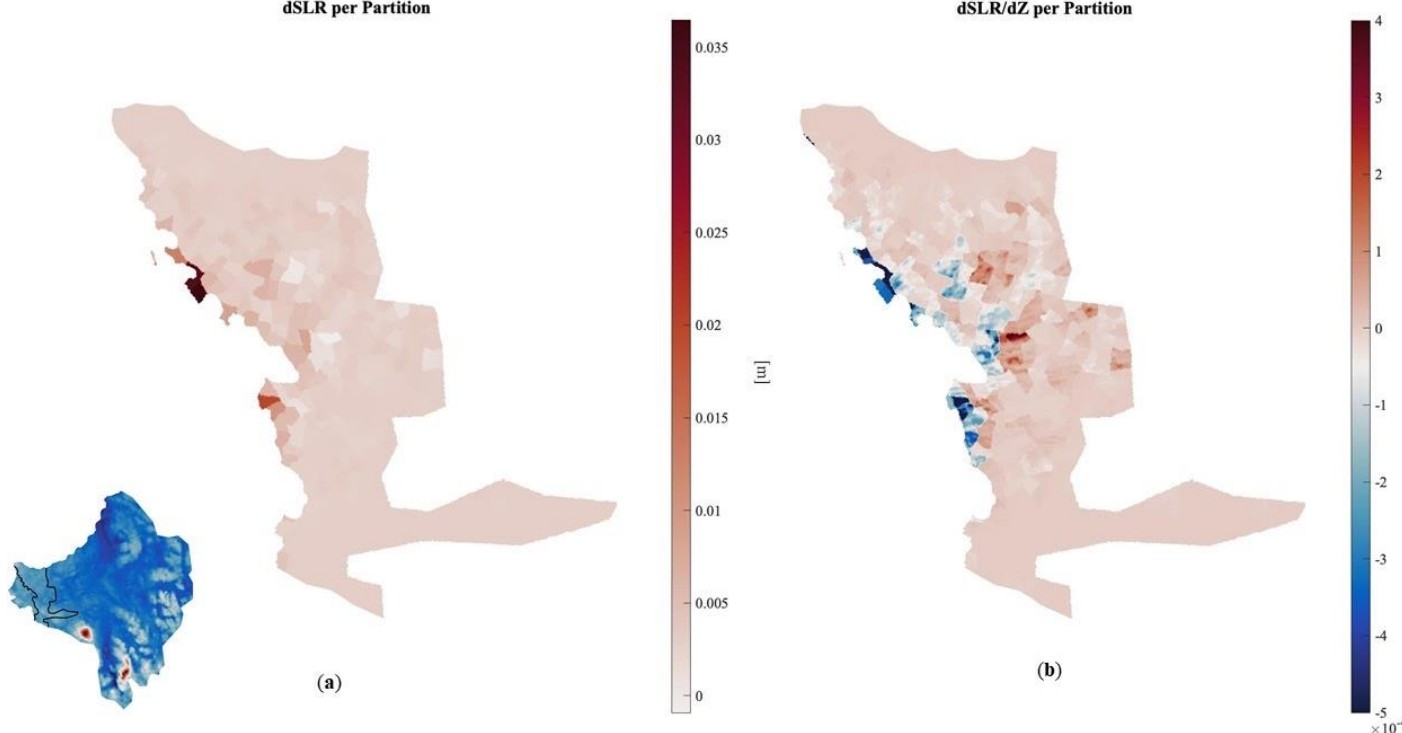

**Figure 6**: **(a)** A plot of the SLR change created by a partition's independent perturbation. **(b)** A plot of the SLR change created by a partition's independent perturbation is divided by the magnitude of the bedrock topography change for each mesh element.

Fig. 6 suggests that model results are most sensitive to changes in bedrock topography closest to the present-day grounding line and local perturbations to bedrock in proximity to the grounding line result in the greatest divergence in modeled SLR contribution. We also note that some locations too are more sensitive to ice thickness losses rather than bedrock volume additions, particularly in the areas where SLR rises in response to a bedrock increase.

We conclude based on these results that the placement of the pinning points matters more than the shape of the pinning points. As observed in Sect. 4.2 (Spatial Resolution Degradation), there was little SLR divergence between different wavelets and spatial resolutions under 2 km spatial representation of the bedrock topography. In Sect. 4.3 (Vertical Resolution Noise Construction), a similar trend was seen as different wavelets and different spatial resolutions resulted in little divergence. Despite using two opposing wavelets, dmey (which focuses on frequency retention) and db2 (which focuses on signal localization), there is little evidence to suggest that either had any role in creating divergent results. The only SLR differences between different wavelet perturbations (arguably negligible) can be attributed to the different amplification multipliers and spatial resolutions.
We also conclude that our results suggest the spatial size of perturbation is negligible compared to the location of perturbation. As stated above, neither of the results of the experiments discussed in Sect. 4.2 (Spatial Resolution Degradation) and Sect. 4.3 (Vertical Resolution Noise Construction) suggests that the spatial resolution of perturbations has any major effect on results.

Fig. 6 similarly does not support any clear conclusion on the effect of spatial perturbation size. Therefore, with limited evidence available to defend such conclusions, we believe further tests will be necessary to examine the model's sensitivity and fully justify this claim.

These sensitivity results suggest the primary bedrock controls of Thwaites exist near the present-day grounding line features.
The sharpness and shape of perturbation do not strongly impact model results, but rather the amplitude and location of the chosen perturbation are more influential bedrock characteristics.

## 6 Discussion

Our experiments aim to quantify the uncertainty in an ice sheet model forward simulation of Thwaites Glacier over a 200-year period of extreme ocean warming. Specifically, we focus on characterizing the bedrock's potential to stop or delay grounding
line retreat and glacial collapse. From these experiments, we also aim to determine how uncertainty in ice sheet model estimates of Thwaites' SLR contribution might be reduced through more accurate and precise data measurements.

Our results suggest that bedrock is an important source of uncertainty in Thwaites Glacier model projections, and in order to constrain projection uncertainty, it is important to minimize present day bedrock data uncertainty when modeling the behavior
of an unstable glacier in response to extreme ocean-driven ice shelf basal melt rates. This is especially the case for our aggressive ocean warming scenario for Thwaites Glacier, which suggests that bedrock topography error could produce an overall spread of 21.9 cm in sea-level contribution (Fig. 2). In order to identify the requirements for spatial and vertical bedrock resolution that would reduce our modeled Thwaites SLR uncertainty to ±2 cm, in accordance with goals outlined by the most recent Decadal Survey for Earth Sciences, we design a unique set of experiments that take advantage of discrete wavelet
decomposition techniques.

Experiments that test spatial resolution requirements suggest that we need bedrock geometry to be measured at a spatial resolution of 2 km or less in order to accurately characterize the bedrock topography pinning points and to minimize ice sheet model uncertainty on 200-year timescales. In particular, we find that at resolution finer than approximately 2 km (at sensitive
glacial regions), estimated SLR contribution converges (Fig. 3). For our assessment of vertical accuracy requirements, we conclude that vertical error needs to be known to an accuracy of ±8 m in order to constrain simulation uncertainty to ±2 cm, especially at major glacial pinning points (Fig. 4) where the grounding line is most likely to be affected by perturbations in





bedrock topography. In fact, we find that our model simulation is highly sensitive to all perturbations in bedrock topography, regardless of the spatial resolutions and shape of the bedrock noise applied.


We also find that, with respect to estimated SLR contribution, perturbation sharpness and shape of bedrock features are negligible in comparison to their amplitude and geographic location. This is because variations in perturbation shape and resolution for both the spatial and vertical resolution experiments result in little divergence in SLR contribution (Fig. 3 and Fig. 4). Results also suggest that vertical resolution results are driven by the amplitude of noise while the spatial resolution

results are affected by the mitigation of high frequency vertical noise (in sensitive glacial regions). These results offer further justification that the amplitude of change in perturbations play a larger role than perturbation shape. In addition to the amplitude of a bedrock feature, the location of perturbation also strongly influences our model estimates of the SLR contribution from Thwaites Glacier. This is because results of our spatial sensitivity experiment suggest that changing the ice thickness locally within a partition does impact our simulation beyond the consequences of altering just the bedrock

topography. We also find that almost all the most influential partitions are within ~50 km the present-day grounding line, suggesting that pinning points are abundant in proximity to Thwaites' rough undersea mountain ridges that persist just upstream of where the glacier is currently grounded.

It is important to restate that our model simulations are forced with highly aggressive basal melting rates. In a perfect world,

we would want to reach bed data at the extreme precision we detail in order to cover all possible situations. However, as the basal melting rate used in our model is extreme, a lower melting rate would yield less spread in SLR contribution and would therefore allocate less precision in spatial and vertical bedrock to reach our ±2 cm SLR goal. In that case, DEMs with worse resolution could be acceptable depending on the modeling data sets and configurations used.

It is also important to note that wavelet decomposition focuses on the amplification of noise that is already present in the bedrock topography map, and therefore it does not create new perturbations. As a result, if pinning points are present in low frequency zones, it may not be sensed by the wavelet tests. We however assume this does not strongly affect our conclusions, as we find that the most influential bedrock regions are almost exclusively near the grounding line where high frequency noise is prevalent. We also recognize some regions may witness a "clipping off" of the top of perturbation shapes in Sect. 4.3 due to

changes in amplitudes going above the maximum bedrock error. Therefore, if SLR has more dependence on the perturbation's curvature or shape than previously suggested, we may lose information regarding the overall shape effect. For spatial resolution degradation experiments (Sect. 4.2), we observe gridding effects when building noise on top of the 200m interpolated DEM. Based on additional sensitivity experiments, we do not believe this creates error in our experiments but it is important to note this could have some effect on the resulting bedrock realizations.




Furthermore, our methods may overrepresent the SLR response as Thwaites Glacier is an active landmass affected by vertical land motion (VLM; Larour et al., 2019). GPS measurements in regions of bedrock uplift have shown trends of VLM in the range of tens of mm/yr which has the potential to affect results (Barletta et al., 2018). However, as the aggression from our high basal melting rate rapidly recedes the grounding line, we do not believe the uplift response has a major effect on our model as pinning points likely would not translate quickly enough to stabilize the glacier. Results likely would be mitigated by this contribution in milder circumstances. As modern uplift couplings become online we hope to further explore its individual SLR contribution.

Model SLR contribution results can have variable results when other bedrock topography DEMs of Thwaites are used. Different DEMs have significant SLR changes as we find that the model is highly sensitive to even slight changes to the mean bedrock elevation. In this case, our results may be biased towards BedMachine. To test this, we did try to compare our model results against a simulation that uses a DEM of Thwaites made from geostatistical data from paleo observations (Mackie and Schroeder, 2019) instead. However, all simulations using this other bedrock realization result in full retreat. We do note that this alternative DEM, however, falls outside of BedMachine's bedrock error range (both above its maximum limit and below its minimum limit) and would therefore not be correctly configured for the sensitive experiments we perform.

Other stress balance models of higher sensitivity to bedrock perturbations may show differing results when compared to simulations that solve the SSA equations. This is due to SSA approximation solutions being less sensitive to small bedrock perturbations than other higher-order stress balance solutions. Therefore, we perhaps may be overestimating our resulting bedrock uncertainty ranges as other stress balance models may produce larger SLR mitigation effects from various bedrock perturbations seen throughout this study.

With respect to biases in our model setup, results could vary based on how the 10 year relaxation period of the control model is performed. Though the relaxation period we run brings the grounding line fairly close to the observed present day Thwaites grounding line, small but potential pinning points may already be passed by the grounding line. The converse too is possible: potential pinning points may not yet be passed by the model grounding line while the observed grounding line is actually already upstream of these pinning points. We also choose to not conduct a new friction inversion for every bedrock configuration due to the high computational price addition to our model. As we believe friction inversion perhaps has the ability to change results, we leave this open question for future investigation as to how much of an impact neglecting the transient friction recalculation may bias our model results.

By design, we use a simplified but highly aggressive basal melt rate approximation in order to capture a wide range of retreat scenarios. Alternative forms of basal melt approximations (interpolation techniques, bedrock altitude dependents, etc) have been observed to change ice sheet model projections and can affect grounding line migration. Furthermore, alternative sliding





laws and sliding coefficients can also influence results depending on the approximation made (i.e. Bulthuis et al., 2019; Alevropoulos-Borrill et al., 2020).

## 7 Conclusion

We conclude that within the currently reported bedrock topography error, it is possible for a simulation of Thwaites Glacier to

remain stable despite aggressive basal melting rates. In particular, we find that positive bedrock perturbations equivalent to the error itself applied throughout the basin promotes stability within the 200-year time period examined (Sect. 3). This suggests that unknown bedrock features within the basin could render Thwaites glacier more stable than would be estimated by a model using BedMachine topography to define its geometry. Finely-resolved bedrock features that may not be captured by current observations should not be neglected in forward simulations. Through the wavelet amplification techniques detailed above

(Sect. 4.1), we determine that accurately resolving bedrock at 2 km and 8 m respectively for spatial and vertical resolutions, especially in proximity to pinning point locations (Sect. 4), would be required to constrain our model uncertainty in 200-year extreme warming projections to within ±2 cm of SLR contribution. We find that the most influential pinning points are located near the present-day grounding line (Sect. 5). Therefore, in order to accommodate finely resolved observations of bedrock topography observations, model meshes should also be at least as fine as 2 km in areas where the grounding line of the glacier

may migrate. Additionally, future bedrock data should aim to limit bedrock uncertainty to ±8 m near the present-day grounding line. Lastly, we conclude that the location and amplitude of a bedrock feature is more important than its sharpness and shape (Sect. 5).

As Thwaites Glacier is unstable, sensitive, and potentially may contribute one of the largest SLRs from the Antarctic Ice Sheet,

our resolution requirements (Sect. 4) may represent a lower bound throughout the continent for bedrock data measurement; more studies may be required to confirm this assumption. This is particularly true, since Thwaites is highly sensitive to retreat in comparison to the rest of the continent.

Our experiments suggest that Thwaites Glacier requires precise data measurements in order to obtain a full projection of its

stability. With the glacier being particularly sensitive to bedrock perturbation amplitude and location, it is highly critical to better constrain uncertainty in simulated SLR contribution that is sourced in bedrock topography. Even small perturbations in highly influential locations can have large implications for ice sheet model estimates of grounding line evolution and projections of future regional contribution to SLR. Overall, improving uncertainty in measurements of bedrock topography will improve certainty in projections of the future evolution of the entire Antarctic Ice Sheet.




# 8 Acknowledgements

The research was carried out at the Jet Propulsion Laboratory, California Institute of Technology, under a contract with the National Aeronautics and Space Administration. Funding was provided by grants from Jet Propulsion Laboratory Research and Technology Development program, the Earth Science and Technology Directorate, Jet Propulsion Laboratory, California Institute of Technology. and from NASA Cryospheric Science, Sea Level Change Team, and Modeling, Analysis and Prediction (MAP) programs. We gratefully acknowledge computational resources and support from the NASA Advanced Supercomputing Division. The authors would like to thank Helene Seroussi for her guidance in the modeling of Thwaites glacier and grounding line model development, Michael Schodlok for his contribution for the Thwaites' Glacier basal melting rates, Mathieu Morlighem for early access to BedMachine data, Dimitris Menemenlis for support with Pleiades Supercomputer allocation, Robert Beauchamp for leading remote sensing simulations of Thwaites bedrock topography, and to Mickey MacKie for early access to her geostatistical bedrock topography data from paleo observations.

# 9 Appendix

## 9.1 Initialization Algorithm

Due to the perturbed bedrock geometry, a series of initialization steps occurs before all runs, we adjust the bed and ice thickness to ensure model stability and avoid instantaneous shock upon restart of the forward model. These steps are as follows:

1. locations at which the ice sheet is ungrounded do not sustain bedrock perturbations to prevent a sudden spontaneous ice sheet advance
2. the new bedrock elevation is set as the base height for all previously grounded features
3. anywhere that the new base goes above the surface has a recalculation to make the surface is at least one meter greater than the new base (i.e. the model minimum ice thickness of 1 meter exists everywhere)
4. we force all grounded areas to be above hydrostatic equilibrium by adding an appropriate amount of ice to unstable element surfaces (allowing a momentary stability to occur within the first few timesteps)
5. locations that have an ice surface less than one meter are set equal to one meter
6. thickness is calculated to be the new surface subtracted by the new base (we implement in this fashion due to ice thickness often being found as a function of known bedrock depth)

## 9.2 Figure Code Combination Legend

Figures 3 and 4 use wavelet amplification techniques to create perturbations of various shapes and amplitudes. The naming techniques in the legends are as follows: 'Wvlt: <Wavelet name> | Lvl: <resolution level> | %: <additional percent amplification>'. The wavelet name is either db2 (Daubechies' 2nd Order Wavelet) or dmey (Discrete Meyer Wavelet), the





resolution level (varying integer between 1-4) is the amount of wavelet decompositions that occur on the low frequency or original image before high frequency amplification occurs, and the additional percent amplification is the amplification multiplier added onto a base multiplier of 1.

**9.3 Tabulated SLR Values for Experiment 1 (Fig. 2): Minimum and Maximum Bedrock Resulting Spreads**

| Time | Minimum Bedrock Realization | Control Bedrock Realization | Maximum Bedrock Realization |
|------|------------------------------|------------------------------|------------------------------|
| **150 yr** | 13.1 cm | 8.7 cm | 2.1 cm |
| **200 yr** | 26.7 cm | 21.1 cm | 4.8 cm |

**10 Notes**

Models are run with ISSM, which is an open-source software. Further information can be found at https://issm.jpl.nasa.gov/. Many figures include colormaps made by Thyng et. al (2016). The MATLAB Wavelet Toolbox is utilized for wavelet analysis. Figure 3 is made using shaded percentile plotting by Onofrey (2020).

**11 Code and Data Availability**

The control bedrock realization from Morlighem et al., 2019 (BedMachine) can be found at https://nsidc.org/data/NSIDC-0756. ISSM software is open source and can be downloaded at https://issm.jpl.nasa.gov/. Scripts and functions used for this project can be found at https://www.dropbox.com/sh/f13f4yvukpjzlyf/AAB2hiIbiWbG2PwHcPADw_Xaa?dl=0.

**12 Author Contribution**

BC led the structure, design, and analysis for all tests performed. NJS was responsible for model setup, DAKOTA sensitivity 705 testing, and analysis of results. LC was responsible for assisting wavelet setup and analysis of the respective results. EL and AK were in charge of guiding and reviewing research progression.

**13 Competing Interests**

The authors declare that they have no conflict of interest.



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
