# Peer review of "Derivation of bedrock topography measurement requirements for the reduction of uncertainty in ice-sheet model projections of Thwaites Glacier"

_The Cryosphere, 2021_

## Author Response (AR1)

Dear Editor and Referees,

On behalf of the co-authors to this paper, thank you for your evaluations, comments, and reviews our paper. Please observe the changes we have made to your requests below and an attached document with all revisions made highlighted in yellow.

Regards,

Blake Castleman

**I    Anonymous Referee #1 Requests & Revisions**

- l42 - perhaps use '0.59' to avoid it being misread (same applies elsewhere, e.g. l146, l257)

  The occurrences of a lack in a leading zero have been fixed.

- l61-2 - 'nearly impossible...using simplified parameterizations' - I think by virtue of them being 'simplified' that implicitly means that not all oceanic processes are being represented, right? I would just say, 'simplified parameterizations does not capture oceanic processes well', or something like that.

  The following sentence has been reworded.

- l62-3 - might be worth mentioning ISMIP6 results here perhaps? Seroussi et al., 2020, Edwards et al., 2021. They show considerable spread, some of which arises from the basal melt rate parameterization. See also IPCC AR6 Chapter 9 (Fox-Kemper et al., 2021).

  The following sentence now mentions Seroussi et al. (2019), Seroussi et al. (2020), Edwards et al. (2021), and Levermann et al. (2020).

- l80 (& 82, 83, 86 etc) - 'errors', or 'uncertainties'? Isn't the point that we don't *know* whether the interpolations are accurate or not? Do we know for sure they are 'wrong'? I realise that 'statistical error' has a specific meaning, but from the perspective of clarity, might be worth considering alternative wording for at least a few instances of this, where appropriate.

  The following uses of the word 'error' have been evaluated and respectively changed to 'uncertainty'.

- l303 - Just a question - is there anything in the surface DEM that could help constrain where 'unknown' bedrock features might be present? Usually we see a surface expression of bedrock rises.

Please view the first comment in the Editor's Requests & Revisions (Sec. IV) for an amendment of the following response.

- l655 - 'highly critical' - surely just 'critical' would do, or just, 'important'?

The following adverb 'highly' has been removed.

- Fig 2 - GL positions are very hard to distinguish, make thicker perhaps, or put a thicker white line underneath each of the coloured lines?

Fig. 2 grounding lines are now 50% thicker.

**II    Anonymous Referee #2 Requests & Revisions**

- L80: plus potentially errors in the interpretation of radiograms -- e.g. misattribution of basal crevasses as bed returns

Potential errors in radiogram interpretation is now discussed within the following sentence.

- L115: Do you specify a minimum ice thickness? i.e. in some simulations do you see the ice shelf thins to a point where the ice shelf front has essentially retreats (but is held in position by the imposed minimum ice thickness)?

The minimum ice thickness of 1 m is now specified on the proceeding line.

- L153: Personally, I would include the leading zero before the decimal.

This occurrence of a lack in a leading zero has been fixed.

- L151: Melt rates extrapolate inland, but later on your mention that there is no melt in partially grounded elements. So is this extrapolation inland for newly ungrounded ice only? Perhaps just clarify this.

Clarification is now provided on the relation of extrapolated melt rates to ungrounded element melt settings.

- Section 4.1: Would it be possible to include a conceptual diagram to explain how the wavelet decomposition works or at least what it ultimately means for adding noise to bed topography? Or perhaps some examples in the SI of the various bed realizations (or difference plots between the perturbed and control beds). This is so that readers who don't follow the details at least can understand what the bed perturbations looks like, aiding their intuitive understanding of the results.

Fig. S4 now visually illustrates an example of how the db2 and dmey amplification processes may change a pinning point. This is mentioned on L255 in the manuscript.

- Section 4.2: What is the assumption being made here about bed roughness under Thwaites? Is the minimum resolution of 2 km that you are proposing only required if high frequency variation in topography actually exists?

Observational evidence has proven the existence of higher resolution features in Thwaites Glacier (e.g. Boon, 2011; Schroeder et al., 2014; Chu et al., 2021). Therefore, we assume that the bedrock topography is not smooth and these features exist on a high amplitude scale. The introduction to Sec. 4.2 has now been revised to reflect this on L319-22.

Please view the second comment in the Editor's Requests & Revisions (Sec. IV) for further discussion and evidence on the following topic.

- L341-2: Because you have already described Fig 3b, this sentence about exceeding 2cm around the 2 km mark seems repetitive. I'm not sure what you mean by 'eventually'…

This sentence has been omitted from the paragraph and the proceeding concluding sentence has been moved to the paragraph above (L340) where the 2 km mark is mentioned.

- L344-5: Similarly, this sentence is confusing – what does compensation mean in this context? Does "before 2 km" mean higher or lower resolution?

This sentence has been omitted from the paragraph (see the revision above).

- L246-349: This is an interesting finding. Can you reiterate the context, i.e. this 2 km is therefore the desired minimum horizontal resolution required for bedrock topography data in order to keep uncertainty in SLR projections due to bedrock data to less than 2 cm.

This has previously been reiterated in the conclusion paragraph (L349-351); the paragraph, however, has now been slightly modified to fit the desired wording.

- L387-8: Perhaps refer back to Experiment 1 here as it seems relevant.

Figure 2 (from Experiment 1) is now referenced in this sentence.

- L404-6: Could you add vertical/horizontal lines on Fig 4 to mark the +/- 2cm SLR and corresponding +/- 8 m vertical res?

Vertical and horizontal lines have now been included for these marks.

- L444: "repeat the bedrock sampling experiments, but this time with perturbation of only the ice shelf basal melting rates". I wasn't entirely sure what you meant by this – from the Fig 5 caption, I think you mean running the control topography (i.e. bedmachine with no errors) for 300 different melt rate perturbations. Could you make this clearer?

This sentence has been reworded to minimize confusion on the setup of this particular sampling experiment.

- L475 and elsewhere: when you provide the range of the distribution, is this a range between certain percentiles, e.g. 5-95% range (given you are presenting PDFs)?

The distribution is not normal, so we have decided to present the numbers as ranges instead. Therefore, the absolute ranges as yielded by the resultant discrete data is presented to the reader. This has now been reworded in the paragraph to provide more clarity.

- L496-7: "locations or strong topographical influence on the grounding line, or pinning points" – I would consider a depression which results in no stable GLs / rapid retreat to also be a strong topographical influence on the GL -- could you find an alternative way of phrasing these locations that are particularly conducive to stable grounding lines?

This has now been altered as to avoid absolute wording of these locations being pinning points. Pinning points and depressions are now both used as examples of topographical influences on grounding line retreat.

- L628-9: Favier et al. 2017 (https://doi.org/10.5194/gmd-12-2255-2019) test a range of melt rate parameterisations and should probably be cited here

This source has now been cited in this sentence and listed among the references.

**III Referee #3 (Helen Ockenden) Requests & Revisions**

- In Experiment 1: Minimum and Maximum bedrock resulting spreads, you make the assumption that the minimum bedrock topography represents the maximum possible retreat and the maximum bedrock topography represents the minimum possible retreat. This is supported by your figures for max bedrock (4.8cm SLR), control (21.1 cm SLR) and min bedrock (26.7 cm SLR). However, beds which contain a mixture of plastic and viscous areas can exhibit behaviour which is outside the range bounded by purely viscous and purely plastic beds. (Koellner et al., 2019, https://doi.org/10.1016/j.epsl.2019.03.026). Do you think that there is a possibility that more retreat could be seen by a scenario between the max bedrock and min bedrock endmembers? The smoothest and roughest beds available within the $3\sigma$ bounds from Bedmachine might provide a different set of extremes? Nias et al (2016, https://doi.org/10.1017/jog.2016.40) might be a useful reference here.

Thank you for your response regarding possible maximum and minimum grounding line retreats within our end members. With this assumption we are technically assuming that the grounding line retreat extremes occur are the minimum and maximum bedrock topography realizations; however, our experience regarding this assumption with Thwaites Glacier has suggested our assumption captures most of the responses of the model (as compared to other glaciers within the Amundsen Sea which cannot use the same assumption). Furthermore, we have analyzed the curves from the DAKOTA runs and have found the minimum and maximum grounding line retreats sit at the extremes of the DAKOTA runs (Sec. 5). This is now addressed within the Discussion (Sec. 6) on L561-65.

- Melt rate obviously has an impact on sea level rise, as you discuss in experiment 5. However, it's not really addressed in this paper whether there is any interplay between the topography sensitivity and the melt rate. If the ocean warmed faster than expected, would a higher melt (ie 2x instead of the max 1.8x you use) mean that a higher topographic resolution is required?

The reviewer poses an important point regarding the interplay of the sensitivity and melt rate. We had previously not covered enough within the discussion of the manuscript to address this. With a melt rate multiplier increase past ×1.8, a decrease in the required resolution threshold is observed. In this case, the glacier is fully collapsed and therefore grounding line retreat has less uncertainty in final positioning. This is now discussed on L597-609 and depicted on Fig. S7.

- You also don't mention at any point the effect of variations in basal slipperiness, or basal sliding law, on the future sea level rise. It would be good to see this addressed in the discussion/future work section.

Basal slipperiness and basal sliding law considerations for future work are now mentioned on L626.

- "Ice sheet model", "Ice sheet instability" and "sea level rise" should technically be written with a hyphen as ice-sheet model, ice-sheet instability and sea-level rise. There are a few instances of this throughout the paper.

The following revisions have been made.

- L17 Upper case G, Glacier

The following revision has been made.

- L50 For basal melting rates, because stochastic evolution is prevalent in ocean circulation, especially beneath ice shelves, melting rates are difficult to accurately observe, model, and predict. >> Basal melting rates are difficult to accurately observe and model, because stochastic evolution is prevalent in ocean circulation, especially under ice shelves.

The following revision has been made.

- L71 Data are technically plural, so 'The data are interpolated'

The following revision has been made.

- L75 It's confusing to say that bedrock topography is directly measurable using remote sensing, because this is not true. We can measure surface elevation using remote sensing, and then use this to interpolate between radar lines to get the bed topography. Instead you could say something along the lines of 'bed topography is directly measurable using ice-penetrating radar (see Holt et al., 2006, Holschuh et al., 2020 etc)(https://doi.org/10.1029/2005GL025561, https://doi.org/10.1130/G46772.1).

The following revision has been made and the additional sources have been added to the reference list.

- L77 Although there are a variety of different digital elevation models, the modelling community has primarily only used Bedmap (v1 and 2) and Bedmachine Antarctica. So this sentence is maybe misleading, and could be removed without altering the paragraph significantly.

The following sentence has been removed from the paragraph.

- L107 Shelfy-Stream (hyphen and capital missing)

The following revision has been made.

- L112 Can you add a sentence clarifying what the modification from Yu et al. (2018) does?

The wording has now been changed to be more precise and indicate the source of our drainage basin definition. This is further detailed on L114-6.

- L114 Is the model of Schlegel et al. (2018) from Schegel et al. (2013)? It seems like you don't need the Schlegel et al (2013) reference here.

The citation has been moved to an earlier point in the sentence to clarify the single point constraint method used.

- L140 Schlegel et al. (2018)

The following revision and other occurrences of incorrect citation formatting have been made.

- L158 Is there a missing 'a' here? "…using a sub-element on a partially floating…"

The following revision has been made.

- L166 I don't think provided is necessary here

The following revision has been made.

- L194 I would change yield to is "The resulting SLR difference between A and B is 21.9cm"

The following revision has been made.

- L196 Figure 2 only shows the grounding line position after 200 years, and not the evolution of the grounding line. Can you confirm here somehow that there is a stabilisation of the grounding line, and that it wouldn't retreat further if the model was allowed to run for longer?

The reviewer is correct, that due to the extreme melt forcing, it is possible that if run for hundreds of years, the grounding line could retreat past the ridge in question in the maximum bedrock scenario. We acknowledge that the use of the word stabilize is misleading in this case. We have reworded the sentence, and we hope that it better represents our meaning, that within the bedrock uncertainty, this ridge may or may not be responsible for temporary stabilization over the time-period investigated here.

- L226 "extreme" not "extremes"

The following revision has been made.

- L227 Thwaites Glacier (insert glacier after Thwaites)

The following revision has been made.

- L298 'produces' not 'produce' (Noise amplification … produces geological landforms)

The following revision has been made.

- L325 I think it's clearer to say 'See Appendix (Sect 9.2) for further details on…'

The following revision has been made on L330 and on L373.

- L330 'Many' rather than 'much' (Many of the perturbations are…)

The following revision has been made.

- L425 You're missing a 'the' here (We use the LHS sampling algorithm)

The following revision has been made.

- L596 The wording here is confusing because Thwaites Glacier is not an active landmass, it's a glacier which sits on top of an active landmass. You could just rephrase this as 'Thwaites glacier is affected by vertical land motion'.

The following revision has been made.

- L601/602 Change 'its' to 'their' (We hope to explore their contribution), because couplings are plural

The following revision has been made.

- L606 Change 'promotes' to 'promote'

The following revision has been made.

- L610 '…would therefore not be correctly configure for the sensitive experiments we perform' >> sensitivity experiments?

The following revision has been made.

- L655 it is highly critical to better constrain uncertainty in simulated SLR contribution that is sourced in bedrock topography >> it is critical to constrain the uncertainty from bed topography in simulated SLR contribution OR it is critical to constrain how much of the uncertainty in simulated SLR contribution is sourced from bedrock topography? Something about this sentence is a little unclear, it maybe needs rewording.

The following sentence has been reworded.

- L668 upper case G, Glacier

The following revision has been made.

- Your reference list is also not consistently formatted, and it's probably faster for you to tidy this up in your files before the typesetting stage

The reference list has been revised and reformatted appropriately.

**IV    Editor (Jan De Rydt) Requests & Revisions**

- In reply to reviewer #1's comment on l303 ("Is there anything in the surface DEM that could help constrain where 'unknown' bedrock features might be present?"): Your answer does not fully recognise the extensive literature on this topic. A seminal paper is https://doi.org/10.1029/2002JB002107, which has sparked many follow-up studies both theoretical and using observations.

Thank you for pointing this out to us. We recognize that we neglected to discuss these important previous studies.  As a result, we amend our response, as indeed, a number of numerically-based studies have been conducted to investigate the relationship between surface features and basal conditions, as well as to examine the physical mechanisms and critical wavelengths responsible for surface expression of bedrock topography (e.g. Gudmundsson et al., 1998; Schoof, 2002; Gudmundsson, 2003). In the past, surface DEM expressions and surface velocity, particularly within ice streams, have been used to infer basal conditions, including simultaneous inversion for basal drag and basal topography (Gudmundsson, 2003).  The aforementioned work being done by M. Morlighem involves incorporating these observations into topographic products that serve as ice sheet model input (such as BedMachine), and this application is still in its infancy.

- In reply to reviewer #2's comment on Section 4.2: "What is the assumption being made here about bed roughness under Thwaites? Is the minimum resolution of 2 km that you are proposing only required if high frequency variation in topography actually exists?". You could potentially strengthen your reply by referring to neighbouring Pine Island Glacier. Several studies on the existence of small-scale topography have been published, e.g. https://doi.org/10.1038/s41467-017-01597-y. And perhaps of further relevance: the interplay between small-scale details of the basal topography and basal traction, both of which control ice dynamics, has been discussed in https://doi.org/10.3389/feart.2018.00033.

Thank you for the suggestions; this evidence helps strengthen our argument for the high-resolution we use. Pine Island Glacier is now further discussed as evidence of Amundsen Sea glaciers containing high frequency noise within bedrock topography (L323-26). In theory, if the features did not exist then we would not need the high resolution to capture all bedrock features. However, past studies show that we would still need model resolution to be at this level to capture grounding line migration accurately (i.e. Seroussi and Morlighem, 2018).

**V  References**

Gudmundsson, G., Raymond, C., & Bindschadler, R.:. The origin and longevity of flow stripes on Antarctic ice streams. Annals of Glaciology, 27, 145-152. doi:10.3189/1998AoG27-1-145-152, 1998.

Gudmundsson, G. Hilmar.: Transmission of Basal Variability to a Glacier Surface. Journal of Geophysical Research: Solid Earth, vol. 108, no. B5, https://doi.org/10.1029/2002jb002107, 2003.

Seroussi, H. and Morlighem, M.: Representation of basal melting at the grounding line in ice flow models, The Cryosphere, 12, 3085–3096, https://doi.org/10.5194/tc-12-3085-2018, 2018.

Schoof, C.: Basal perturbations under ice streams: Form drag and surface expression. Journal of Glaciology, 48(162), 407-416. doi:10.3189/172756502781831269, 2002.

---

## Author Response (AR2)

[revised manuscript text omitted]